# Comprehensive Accounting for REDD+ Programs: A Pragmatic Approach as Exemplified in Guyana

**Katherine M. Goslee [1],\*, Timothy R. H. Pearson [1]** 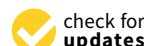 **, Blanca Bernal [1], Sophia L. Simon [1] and Hansrajie Sukhdeo [2]**

[1]    Winrock International, Arlington, VA 22202, USA; tpearson@winrock.org (T.R.H.P.);
       blanca.bernal@winrock.org (B.B.); sophia.simon@winrock.org (S.L.S.)
[2]    Guyana Forestry Commission, Georgetown, Guyana; hans.sukhdeo@gmail.com
\*    Correspondence: kgoslee@winrock.org

**Abstract:** Completeness is an important element for Reducing Emissions from Deforestation and forest Degradation (REDD+) accounting to ensure transparency and accountability. However, including a full accounting for all emission sources in a REDD+ program is often resource-intensive and cost-prohibitive, especially considering that some emission sources comprise far less than 10% of total emissions and are thus considered insignificant according to Intergovernmental Panel on Climate Change (IPCC) guidance. This is evident in forest reference emission level (FREL)/forest reference level (FRL) submissions to the United Nations Framework Convention on Climate Change (UNFCCC). Of the 50 countries that have submitted FRELs to date, only half of them include degradation in their FRELs even though degradation is often a significant source of emissions. Half of the countries that do include degradation use satellite imagery without necessarily specifying degrading activities or separating anthropogenic activities. Guyana provides an example of an approach that enables inclusion of all emission sources while considering the significance of each when developing an accounting approach. Since submitting its FREL in 2014, Guyana has made stepwise improvements to its emission estimates so that the country is now able to report on all deforestation and degradation activities resulting in emissions, whether significant or not. Based on the example of Guyana's efforts, the authors recommend a simple approach to move towards complete accounting in a cost-effective manner. This approach can be scaled to other countries with other activities that results in greenhouse gas emissions from deforestation and forest degradation. Such complete accounting allows for higher accountability in REDD+ systems and can lead to greater effectiveness in reducing emissions.

**Keywords:** REDD+; greenhouse gases; emissions accounting; deforestation; forest degradation

---

## 1. Introduction

Global emission reduction programs, such as the Reducing Emissions from Deforestation and forest Degradation (REDD+) and Nationally Determined Contributions (NDCs) programs, include a commitment to accurately assess changes in forest carbon stocks and associated emissions [1]. Sound and transparent accounting and accountability are necessary to demonstrate commitments to reduce greenhouse gas (GHG) emissions and to justify associated financial transactions [2–5].

Completeness is an important tenet of any forest carbon inventory [6]. This is especially true in a REDD+ context where exclusion, for example, of specific drivers of land cover change or specific pools/gases could substantially change the estimate of net emissions. Inventory completeness includes spatial completeness of forests within the reporting country, completeness in the direct drivers of deforestation or forest degradation, and completeness in the carbon pools and greenhouse gases tracked through time. However, absolute completeness can result in prohibitive costs, especially where

consistent high-quality inventory methods are used, and limited resources exist for inventories [7]. At the same time, not all geographic areas in a country have a significant potential for emissions, not all drivers are responsible for significant emissions, and certain pools may represent only a very small proportion of total stocks [2,4].

Because it is not feasible to develop a perfect inventory approach with full and complete accounting and low uncertainty, countries often sacrifice completeness, excluding important sources, pools, or drivers when a full inventory approach is not available. However, it is possible and acceptable to employ an inventory and accounting approach that is pragmatic and balances the methods used, uncertainty in estimates, significance of emission sources, and costs. Across any inventory, simplifications and approximations can be valid to retain completeness while constraining costs, especially when accounting for emission sources that represent a small proportion of total emissions.

Aspects of such a practical approach exist in international measurement and accounting methods such as Tier 1, Tier 2, and Tier 3 accounting under the Intergovernmental Panel on Climate Change's Guidance for National Inventories [8], or various offered accounting options under the World Bank's Carbon Fund Methodological Framework [9]. However, there are shortcomings in these methods. The Carbon Fund, for example, suggests the use of "proxies" where better methods do not exist or are cost-prohibitive, but these are not tied to the size of the potential greenhouse gas emission and are not designed as scientific, pragmatic, long-term solutions.

Here we use country forest reference emission level (FREL)/forest reference level (FRL) submissions to the United Nations Framework Convention on Climate Change (UNFCCC) to examine the completeness of REDD+ programs. We use the example of Guyana to demonstrate a pragmatic approach where completeness can be achieved in a manner that balances the significance of emission sources with the cost and precision of emission estimates.

## 2. Forest Carbon Accounting

### 2.1. Basic Overview

Forest carbon accounting addresses emissions and removals from different direct drivers, carbon pools, and greenhouse gases, establishing a baseline of forest GHG emissions and removals to which current and future levels can be compared. This initial benchmark is vital in assuring additionality and allowing participation in global carbon markets [10]. Under REDD+, all developing countries seeking financial compensation must develop a forest reference level (FRL) and/or forest reference emission level (FREL), required by UNFCCC Decision 1/CP.16 [11]. The FREL/FRL provides a historical reference against which future emissions can be compared, typically the historical average of recent emissions with or without adjustment for country circumstances or projection into the future. The FREL/FRL is compared with actual emissions that are estimated under a country's monitoring following implementation of REDD+. The difference between the FREL/FRL and actual emissions defines the reduction in emissions that can be attributed to a country's REDD+ activities.

FREL/FRLs estimate emissions from one or more of the five REDD+ activities defined by the Cancun Agreement, including deforestation, degradation, conservation, sustainable forest management, and forest enhancements (afforestation and reforestation) [1]. Activity data (AD) can be sourced from satellite imagery, national, regional, project or plantation records, from national forest inventories (NFIs), or models. Emissions from degradation may be delineated by drivers (e.g., woodfuel collection, logging, grazing, or fire, among other) or reported as a whole. FREL/FRLs also consider different carbon pools, accounting for aboveground biomass (AGB) at a minimum, as well as belowground biomass (BGB), deadwood, litter, and soil organic carbon where relevant and feasible. Carbon dioxide is the dominant gas in forest GHG accounting, but some estimates also comprise methane and nitrous oxide emitted from fires, decomposition, fertilizer use or flooded soils, among other sources. Developing emission factors (EFs) to account for the various activities mentioned above, pools and gases often rely on NFIs, global datasets, forest sampling exercises, or Intergovernmental Panel on

Climate Change (IPCC) defaults. Both EFs and AD have an inherent associated uncertainty resulting from the quality and completeness of available data.

Developing a FREL/FRL and estimates of annual emissions under REDD+ can present a number of challenges and countries have struggled to produce comprehensive national forest accounting due to limited capacity and high costs associated with detailed accounting [10]. UNFCCC negotiations have accepted the barrier these challenges present and as a result allow countries to take a stepwise approach to developing their FREL/FRL and National Forest Monitoring System, and implement an initial REDD+ program that is limited to those elements that can be easily addressed, then improving methods and expanding data, area, and activities addressed over time.

## 2.2. Status of Global Efforts

Fifty countries have submitted FRELs to the UNFCCC to date (September 2020; see Supplementary Materials), with a varying level of inclusion of activities. All countries except India include deforestation in their FREL submission. About half of the countries include forest degradation emissions (Figure 1), estimated in two different methods. Twenty-four percent of submissions use satellite imagery to assess canopy cover change that does not result in deforestation. This approach does not identify specific degrading activities and may not separate anthropogenic from non-anthropogenic causes of decreases in canopy cover. The other 24% of countries that include degradation estimate the emissions of specific degrading activities. Half of the FREL submissions include emission removals from enhancements achieved with afforestation and/or reforestation efforts, while only 12% include estimates of increases in carbon stocks from forests remaining forests. Of the 50 countries, five specifically include a sustainable forest management activity in their FREL (Figure 1).

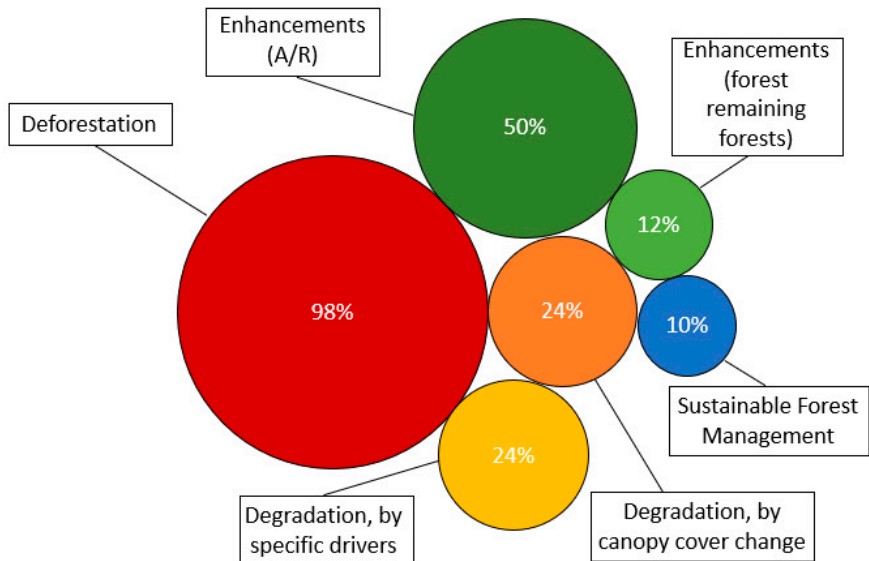

**Figure 1.** Proportion (%) of forest reference emission level/reference level (FREL/FRL) submissions to the UNFCCC that include the Reducing Emissions from Deforestation and forest Degradation (REDD+) activities of deforestation (red), degradation (orange), enhancements (green), and sustainable forest management (blue). The size of the bubbles is proportional to the size of the sample. Degradation is represented by two mutually exclusive bubbles—i.e., countries either report degradation by driver activity or by canopy cover change.

Estimating uncertainty, either of individual activities or the full country FREL, provides insights on the accuracy of the emission and removal estimates and the range within which the actual value can be confidently found [12], and increases the transparency of REDD+ [13]. The uncertainty of specific activities and/or the total FREL has been more frequently reported in recent FREL submissions [13],

yet to date it has been reported in fewer than two-thirds of the submissions from the 50 countries (Figure 2). In some cases, only the total FREL uncertainty was included, without indicating individual activities' uncertainty (e.g., Indonesia and Kenya). In other cases (e.g., Myanmar), only the uncertainty of some of the REDD+ activities included in the FREL was reported. The lack of consistency in reporting uncertainty for AD, EF, and aggregate activity or FREL uncertainty suggests that understanding error propagation and combined uncertainty might be a barrier to accounting completeness. Of the countries that report uncertainty, about half indicate that it is higher than 20% of the FREL emissions, with only 16% reporting uncertainties below 10% of the mean (Figure 2). Because of the different methods applied to estimate uncertainty across submitted FRELs, results cannot be consistently compared, and thus it is unclear if the differences in uncertainty are due to methodological approaches or to data accuracy.

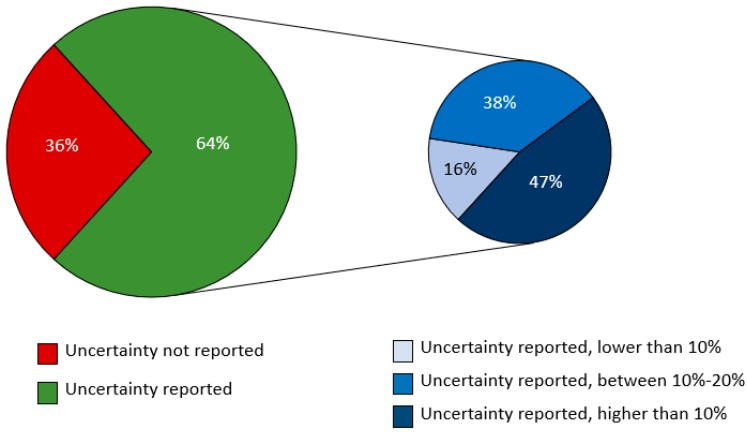

**Figure 2.** Proportion (%) of FREL submissions to the UNFCCC that include uncertainty estimates (**left**); for those that report uncertainty, the proportion (%) of submitted FRELs with uncertainty below 10%, between 10% and 20%, and above 20% are indicated on the right pie.

In addition to the REDD+ activities included and the assessment of accuracy, the carbon pools and gases included are also an indication of FREL completeness. Over 40% of the FRELs report living biomass carbon stocks only (i.e., above- and belowground biomass; Figure 3A), and only eight countries of the 50 total include living biomass, dead biomass, and soil carbon pools. Soil carbon is often challenging to measure and monitor, due to the need for field and laboratory equipment to determine soil density and carbon concentration. Some countries, however, include soil carbon despite the inclusion of only some biomass carbon pools. This is the case for Colombia, which includes soil carbon and living biomass in its FREL, while Indonesia includes soil carbon and aboveground biomass. Furthermore, while all countries assess $CO_2$ emissions, less than 20% also include non-$CO_2$ emissions (methane and nitrous oxide; Figure 3B), typically associated with fire activities that result in deforestation and/or forest degradation and have a higher global warming potential than that of $CO_2$ [14].

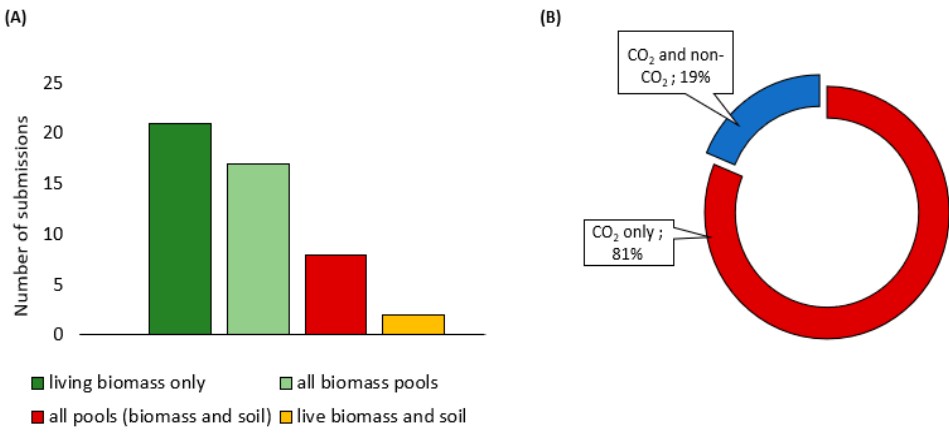

**Figure 3.** (**A**) Number of FREL submissions to the UNFCCC that include only living biomass carbon pools, all biomass (i.e., living and dead) carbon pools, all carbon pools (i.e., all biomass and soil), or only living biomass and soil carbon pools. (**B**) Proportion (%) of FREL submissions to the UNFCCC that include only $CO_2$ gases (red) and $CO_2$ and non-$CO_2$ gases (blue).

The average contribution of degradation to the total country gross emissions in the FRELs submitted is 41.8%, ranging between 5.9% and 94.9% (Supplementary Materials). Only two of these countries (Ghana and Guyana) include estimates of legal and illegal logging in their degradation reference level. Pearson et al. [15] estimated that forest degradation emissions in developing countries, although highly variable depending on the region, could be equivalent to one-third of those from deforestation on average, with a third of the countries assessed owing more than half of their forest emissions to degradation. Of these degradation emissions, half were attributed to timber harvest only, equivalent to 12% of deforestation emissions [16]. This means that half of the countries that have submitted a FREL to the UNFCCC are likely missing a significant portion of their forest emissions by not accounting for forest degradation. While some countries (e.g., Zambia or Ecuador, among others) acknowledge in their FREL that degradation might be a significant unaccounted source of emissions, the most common justification reported for not including this activity is the country's concern regarding its ability to obtain reliable data [15].

While accounting for any form of degradation is challenging, estimating degradation from illegal logging is even harder. However, illegal logging can be a significant source of emissions that are not accounted for, particularly in tropical countries where logging is a common practice. Contreras-Hermosilla et al. [17] estimated that illegal logging in tropical countries could be equivalent to 40% of all tropical logging, with countries such as Brazil as high as 72% [18]. Forest fires can also be an important source of emissions, estimated to be responsible, on average, for 17% of global degradation emissions [15]. Although 19% of the countries accounted for non-$CO_2$ emissions produced by fire (Figure 3B), only two countries (Bhutan and Ghana) report an estimate of the contribution of these emissions to the total forest emissions.

## 3. REDD+ in Guyana

### 3.1. Overview

Guyana offers an example of an application with a pragmatic approach to completeness in REDD+ reporting. Guyana is a leading REDD+ country, with a forest area of 18.4 million ha in 2020, equivalent to 97% of the national area [19]. Guyana has a diverse range of drivers that impact its forests including agriculture, mining, timber harvest, and infrastructure [20]. Some of the drivers are highly significant while others represent a fraction of a percent of total emissions, likewise for forest carbon pools and post-deforestation carbon stocks.

The Governments of Guyana and Norway entered into a partnership in 2009 with the intent of developing a replicable model for aligning REDD+ implementation with country development goals. With funding from Norway, Guyana has developed a Forest Carbon Monitoring System (FCMS), a Monitoring, Reporting, and Verification System (MRVS), and a REDD+ Forest Reference Emission Level (FREL) submitted to the UNFCCC in 2014 [21]. The Guyana Forestry Commission estimated historical forest emissions from 1990 to 2010 and has been monitoring and reporting annual emissions since 2010.

Guyana's initial efforts to develop its FREL and monitor annual emissions focused only on the major sources of emissions—deforestation and forest degradation from logging—and addressed only the area of the country with the highest potential to experience land cover change. The country has made stepwise improvements to the emission estimates since 2010, with areas and emission sources added or improved as possible. As of 2018, emissions are estimated across the entire country from all major and minor emissions sources in Guyana: deforestation due to mining, infrastructure, settlements, and agriculture, and forest degradation associated with timber harvest, mining, infrastructure (roads), and fire [20]. As a high forest/low deforestation country, Guyana does not engage in activities that result in substantial carbon removal, and so enhancements are not included in their REDD+ system. However, enhancements represent an area where pragmatic approaches would be very relevant with respect to determination of the annual areas of new forest regeneration and the associated removal factors.

Given the very small proportion of certain emission sources, such as roads associated with mining or timber harvest, or the fine debris on the forest floor, having a complex and costly accounting system would be a poor use of resources. These situations represent important examples of where simplified approaches to minor emission sources can function as a pragmatic answer to completeness paired with constrained costs.

The national forest inventory in Guyana seeks to accurately account all greenhouse gas emissions and sequestration but in a pragmatic and sustainable manner.

### 3.2. Deforestation and Forest Degradation GHG Emissions in Guyana

Analyses associated with Guyana's National Forest Monitoring System estimated an area of approximately 9038 ha deforested per year on average between October 2010 and December 2018, with a high of 14,655 ha in 2012 and a low of 7912 ha in 2011 [20]. This total is dominated by deforestation for mining, including infrastructure for mining, which together account for 87% of all deforestation. Agriculture accounts for 4.1% of deforestation, while fire accounts for 4.4% and forestry infrastructure accounts for 2.5%. Settlements and other infrastructures account for less than 2% each.

Forest degradation is dominated by timber harvest with 627 million cubic meters harvested and extracted per year on average between 2010 and 2016 (high of 760 million in 2014, low of 501 million in 2016).

Guyana's emissions were an estimated 11.4 million tons of carbon dioxide equivalent (Table 1). Deforestation from mining and degradation from logging were the largest sources of emissions, accounting for a combined 79.5% of all emissions. All sources of deforestation accounted for 83% of emissions. Degradation from sources other than timber harvest accounted for less than 1% of total emissions.

**Table 1.** 2018 emission estimates for all sources under Guyana's REDD+ system.

| Activity | Emissions Estimate (×1000 t $CO_2$e) | Percent (%) of Total Emissions |
|---|---|---|
| **Deforestation total** | **9514** | **83.4%** |
| **Mining** | 7249 | 63.5% |
| **Mining infrastructure** | 719 | 6.3% |
| **Agriculture** | 565 | 5.0% |
| **Fire/biomass burning** | 532 | 4.7% |
| **Forestry infrastructure** | 372 | 3.3% |
| **Infrastructure (primary roads)** | 70 | 0.6% |
| **Settlements** | 7 | 0.1% |
| **Forest degradation total** | **1900** | **16.6%** |
| **Timber harvest** | 1831 | 16.0% |
| **Mining degradation** | 56 | 0.5% |
| **Illegal logging** | 11 | 0.1% |
| **Infrastructure degradation** [1] | 2 | 0.02% |
| **Total emissions** | **11,414** | **100.0%** |

[1] Infrastructure degradation is estimated based on percent of total emissions derived from 2016 data.

### 3.3. IPCC Key Category Analysis

Key Category Analyses (KCAs) is recommended by the IPCC Good Practice Guidance [22] and the IPCC Guidelines [8] to inform priorities for GHG inventory preparation and improvement. KCA helps identify the most significant sources and sinks of emissions as well as those categories and subcategories with a trend that differs significantly from the overall trend of the inventory, allowing prioritizing efforts to improve inventory quality and decrease uncertainty of estimates.

The activities in Guyana's forest sector that have the most significant impact on GHG emissions have well established, highly accurate, and cost-effective approaches to estimating GHG emissions. These activities are deforestation (from mining, infrastructure, agriculture, and settlements) and forest degradation from selective logging. GHG emissions from deforestation were assessed using the stock-change approach [8]. GHG emissions from timber harvest degradation were assessed using the IPCC gain–loss approach [16].

The remaining sources of emissions from forest use in Guyana include fire and degradation associated with mining and infrastructure. Emissions from these sources are estimated using simplified methods that enable inclusion without incurring inordinate resources.

Deforestation from fire is accounted based on a stock-change approach using sampling to develop activity data and IPCC defaults for emission factors. Activity data are derived by estimating the total area of deforestation from fires using a sample-based approach, with satellite imagery used to identify degradation within select areas, and then extrapolated out to the whole country [20]. The emission factor applied is calculated using equation 2.27 in Chapter 2 of Volume 4 of the IPCC 2006 AFOLU (Agriculture, Forestry, and Other Land Uses) report [8].

Degradation associated with mining in Guyana is the result of damage to trees in the area surrounding mines, resulting in a loss of canopy and corresponding emissions. Forests adjacent to mines commonly experience some level of activity, such as individual tree removal in the development of mining camps, resulting in a loss of carbon stocks, although forest cover remains. While the emissions resulting from mining degradation are relatively small, they must be included in REDD+ accounting to ensure completeness in reporting [23]. The method Guyana uses for forest degradation associated with mining is to establish a buffer zone of an established width (100 m, based on an analysis conducted for Guyana Forestry Commission) and develop an emission factor for the entirety of the buffer zone. Development of activity data therefore requires calculating the total area within 100 m buffers around all new areas of mining deforestation in a given year. Brown et al. [23,24] assessed the loss of trees in the forests surrounding mines by establishing 100 m transect plots originating in mines.

Tree mortality was identified along the transects, and the commensurate carbon loss was estimated. All carbon loss that was not a result of natural tree mortality was considered an emission from mining and was used to develop the emission factor.

　　Forest degradation also occurs around infrastructure in Guyana as roads open access both to adverse environmental conditions and to human encroachment. Analysis based on limited high-resolution imagery sampling showed that forest degradation around infrastructure represents a fraction of the emissions caused by mining forest degradation which itself is highly insignificant in the key category analysis (Table 1). Guyana has no field data associated with infrastructure forest degradation but given the low significance has elected to use the method and factor developed for mining to allow the inclusion of this small additional emission source. Given the lack of a resident population around infrastructure as opposed to mines, this will lead to conservative values but will incur no additional costs.

　　Guyana includes all relevant carbon pools in its REDD+ reporting, including biomass from live trees—both above and belowground, saplings, standing and lying deadwood, and litter, and soil carbon (Table 2). A Forest Carbon Monitoring System is implemented in Guyana, under which nested plots have been established across the country and field measurements are collected on a cyclical basis.

**Table 2.** Pools included in Guyana's REDD+ Forest Carbon Monitoring System.

| Pool | Approach | Source |
|------|----------|--------|
| **Aboveground biomass** | Calculated as a function of wood density and diameter at breast height, using equation for moist tropical forests | [25] |
| **Belowground biomass** | Ratio of aboveground biomass | [26] |
| **Saplings** | One-time destructive sampling to develop an average dry weight per sapling and count of saplings at each plot | [2] |
| **Standing dead wood** | Calculated as a function of volume and density of dead wood by class | [2] |
| **Lying dead wood** | Line-intersect method, using density of dead wood by class | [27] |
| **Litter** | Litter "clip plots", with samples oven-dried | [2] |
| **Soil carbon** | Soil carbon sampling with corer and bulk density measurements from soil pit | [28] |

　　While some of the included pools are minor in contribution to emissions, their inclusion is considered pragmatic because required measurements are either one time, as in the case of destructive sampling for saplings, or take a negligible amount of time once a field crew is at the plot needed to conduct measurements for the major pool of aboveground biomass, given that in Guyana, the majority of time needed for field work is in travelling to the plots. Additionally, the schedule of the field inventory in Guyana is designed to ensure that data are collected annually, both to enable updates to emission factors as needed and to ensure that crews are well-versed with measurement protocols, but on a rolling cycle so that while all plot data are replaced over 20 years, only 5% of plots are remeasured every year.

　　Guyana includes all greenhouse gases in its REDD+ reporting. Carbon dioxide is reported for all activities. Methane and nitrous oxide are only relevant to deforestation and forest degradation associated with fires. These gases are included through IPCC emission factors calculated using equation

2.27 in Chapter 2 of Volume 4 of the IPCC 2006 AFOLU report [8], with no additional costs beyond the inclusion of only $CO_2$ emissions.

The Forest Carbon Monitoring System of Guyana has always included estimates of uncertainty including in the FREL submitted by the country to the UNFCCC. Uncertainties are based on sampling and measurement uncertainty, with the relevant data collected as an integral part of the national inventory system, as well as uncertainty in remote sensing, which is gathered in the annual MRVS report and the accuracy assessment (e.g., [20,29]. Current methods for estimating annual uncertainty are based on Tier 1 propagation of errors, but Guyana is moving toward an enhanced approach using Monte Carlo simulations [8]. An approach to Monte Carlo is being developed that will allow annual automated implementation, to achieve more reliable uncertainty estimates without incurring excessive costs.

A cost-assessment has not been conducted on the pragmatic approaches used in Guyana, largely because the full detailed methods have never been implemented in full as part of the work of the REDD+ Secretariat in Guyana. This lack of implementation was strongly driven by the very high associated costs. For example, the need for annual purchase of high-resolution data and substantial annual time requirements for expert image interpretation for detailed diffuse degradation assessments assures costs of a full detailed analysis will be many times higher than the very limited costs of the pragmatic approaches. Thus, every case taking a pragmatic approach achieves significant cost savings for emission sources that are of low significance.

These simplified methods to include minor sources of emissions enable an annual investment of minimal resources, while ensuring that all relevant emission sources are included in Guyana's FREL and annual emissions reporting. This both ensures completeness and enables annual tracking to determine whether there is a significant increase in emissions such that a more rigorous method of emissions accounting should be pursued.

We can use this example of Guyana to identify high level potential differences between a pragmatic approach and a detailed approach, illustrating the differences in time, resources, capacity and financial cost that can arise between the approaches. Table 3 summarizes potential options for pragmatic approaches versus detailed approaches for activities, pools, and gases commonly excluded by countries (as discussed in Section 2). The cost and effort differential is self-evident and illustrates that comprehensive accounting can be followed in a manner that is pragmatic and does not excessively inflate inventory costs.

**Table 3.** Illustrative summary of key differences (with implied costs) between a pragmatic and a detailed approach to accounting for activities, pools and gases that are often excluded from REDD+ systems. (EF = emission factor; AD = activity data).

|  |  | **Potential Pragmatic Approach** | **Detailed Approach** |
|---|---|---|---|
| **Inclusion of diffuse degradation sources †** | EF | Limited fieldwork and analysis | Fieldwork and analysis |
|  | AD | Buffers applied around new areas of deforestation (minimal additional analysis) | Annual high-resolution imagery with processing and analysis |
| **Inclusion of emissions from fire** | EF | IPCC defaults | In-country fieldwork to determine fire emission factors |
|  | AD | Use of global fire databases (e.g., [30]) | Country-specific remote sensing analysis |

**Table 3.** *Cont.*

| | | Focus on *non-forest to forest* | *Non-forest to forest* AND *Enhancement in forest remaining forest* |
|---|---|---|---|
| **Inclusion of enhancement activities** | EF | IPCC and literature (e.g. [31]) for removal factors. | Extensive scientific studies to establish country-specific removal factors |
| | AD | National statistics on forest plantations and restoration, or remote sensing | Remote sensing (high resolution analyses needed for enhancements in forest remaining forest) |
| **Comprehensive inclusion of carbon pools** | | Use of IPCC defaults for dead organic matter and soil organic matter pools | Extensive fieldwork and laboratory analyses |
| **Comprehensive inclusion of gases** | | Use of IPCC defaults for $CH_4$ and $N_2O$ | Extensive scientific studies to establish country-specific factors |
| **Uncertainty** | | Low uncertainty for activities, pools and gases that contribute significantly to total emissions and higher uncertainty for remaining activities, pools and gases | Low uncertainty for all activities, pools and gases |

[†] Diffuse degradation defined as degradation occurring in an unpredictable pattern in areas immediately surrounding deforestation [24].

## 4. Recommendations for Comprehensive and Cost-Effective REDD+ Programs

The approach taken by Guyana has the potential to be replicated in other countries that currently do not account for all deforestation and degradation activities individually. We recommend the following steps to move towards complete accounting in a cost-effective manner:

1.  Conduct a key category analysis (KCA) to assess significance of emission sources: While a KCA itself can be resource-intensive if all emissions are accounted with a relatively high level of certainty, there are alternative approaches that can be employed with initial approximate estimates of emission levels. Countries should develop order of magnitude estimates of emissions from all sources based on the best currently existing data, which in some cases may include global datasets and/or default values. From these estimates, it will be possible to establish which sources are likely to be significant and which are minor.

2.  Prioritize and pursue complete accounting for significant sources: For any source of emission, pool or gas that is expected to comprise more than 10% of total emissions, countries should apply a rigorous approach that allows full accounting with a low level of uncertainty.

3.  Engage in simplified accounting for insignificant sources: For the remaining sources of emissions, pools or gases, those comprising less than 10% of total emissions, countries should identify an appropriate cost-effective method to estimate emissions. This may entail use of Tier 1 defaults, application of appropriate literature values, or development of a simplified and cost-effective method to estimate emissions.

This approach would enable countries to estimate all emission sources and would allow for better understanding of possible options to reduce emissions, improve forest/natural resources management, and meet emission reduction targets more effectively even under limited budgets.

**Supplementary Materials:** The following are available online at http://www.mdpi.com/1999-4907/11/12/1265/s1. TableS1: List of activities included in the FRELs submitted by the countries to the UNFCCC, as of September 2020, and contribution (%) of degradation to gross forest emissions in countries that include degradation reference levels, Table S2: Uncertainty, carbon pools, and gases included in the FRELs submitted by countries to the UNFCCC, as of September 2020.

**Author Contributions:** T.R.H.P. and K.M.G. conceived of this paper. K.M.G. led the analysis of Guyana's emission estimates. B.B. and S.L.S. assisted with research. All four authors contributed to writing. H.S. was integral to data collection and analysis. All authors have read and agreed to the published version of the manuscript.

**Funding:** The underlying work that led to this paper was funded by the Guyana Forestry Commission.

**Acknowledgments:** Pradeepa Bholanath and Nasheta Dewnath provided critical support and input to the development of the concepts in this paper. Pete Watt, Danny Donoghue, and Towana Smartt were integral in the development of emissions estimates, providing estimates for area change.

**Conflicts of Interest:** The authors declare no conflict of interest.

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
