# Peer review of "Comprehensive Accounting for REDD+ Programs: A Pragmatic Approach as Exemplified in Guyana"

_forests, doi:10.3390/f11121265_

Round 1

Reviewer 1 Report

This is a very important paper that shares the country's experience in achieving complete carbon accounting in a cost-effective manner. 

There are some areas that I believe could make this paper better:

  1. Readers would benefit if the authors could clearly explain the challenges that countries face in submitting FREL in addition to a few mentioned in the manuscript. For example - sampling design to account for different forest types and land uses across the country; Uncertainty estimates and their sources; and error propagation. Accuracy and confidence in different estimates are key. 
  2. From 1, explain how Guyana was able to overcome these challenges using a step-wise approach. It will be good to document for example how Guyana's sampling design, error estimates and uncertainty. 
  3. Make clear recommendations. 

Specific comments 

Introduction - sentence in lines 51-52. The authors could elaborate a bit more by explaining some elements that are valid to retain completeness, accuracy and cost-effectiveness.

Forest carbon accounting - I would suggest a clear definition of FREL/FRL and why is it so important. Lines 106-120 talk about uncertainty superficially but no further mentions how uncertainty has been estimated in the context of Guyana in  3. REDD+ in Guyana.

Authors owe readers more details on Guyana approach and need to strengthen their recommendations. 

Author Response

Thank you for your comments. Please see attachment for our responses to all reviewers.

Reviewer 2 Report

This perspective provides a well-written assessment of countries' deficiencies in their respective Forest Reference Level/Forest Emissions Reference Level for REDD+ under UNFCCC guidance. It focuses on Guyana's reporting method as a potential template for other countries to fill gaps and reduce uncertainties in their FRL/FREL standards in a purported cost-effective manner.

Major comments:

For completeness of the perspective of encouraging other countries to adopt 'Guyana's pragmatic' reporting approach, I would highly recommend the inclusion of a cost-analysis that contrasts Guyana's approach with that of different methods and associated tradeoffs related to accuracy and precision. Some of the baseline data related to degradation (specifically logging) for other countries FREL has also recently become available (see Ellis et al. 2019/https://doi.org/10.1016/j.foreco.2019.02.004).

Another missing piece of the narrative is related to enhancements, for which it seems Guyana is not currently reporting under its FREL (Table S1). Maybe the authors can highlight opportunities and methods to account for these activities that lead to the inclusion of enhancements. A long-term perspective for continuous forest carbon stock assessment based on rapidly emerging tools and methods, especially those related to active sensors (radar, LiDAR), would be informative (i.e., would is the next step for Guyana's reporting system). This inclusion will allow countries to potentially leapfrog some of the incremental steps Guyana has adopted to build out their monitoring and reporting systems.

Minor comments

Error related to reference formatting : Line 111; Line 118; Line 127; Line 134; Line 160; Line 199

Line 112: 'uncertainty was included, without'

Line 161: 'report an estimate of the'

Author Response

(The authors gave the same response as above.)

Round 2

Reviewer 2 Report

Unfortunately the major flaw of this perpective remains with the mismatch between the title 'Complete accounting for REDD+ programs: A pragmatic approach as exemplified in Guyana' (overall conclusions) and the actual critical data needed to make such a claim - that is the pros and cons of Guyana's reporting method compared to other countries' approach to their accounting systems for their FREL under UNFCCC requirements. I am actually a little more alarmed to learn now in the author's response that the 'full detailed methods have not been implemented' - then how do the author's arrive at their recommendations/conclusions on an incomplete system?

I however note their is an institutional collaborator on the author's list who might be able to fill these gaps and help provide at least a lessons learnt approach on challenges, and local costs - human, financial, technological, other. These should then be compared to other reporting approaches to establish the connective tissue for the intent of the perspective.

Author Response

We appreciate the comments. Please see responses in attachment.
